# Variance Regularized Counterfactual Risk Minimization via Variational Divergence Minimization

## Abstract

Off-policy learning, the task of evaluating and improving policies using historic data collected from a logging policy, is important because on-policy evaluation is usually expensive and has adverse impacts. One of the major challenge of off-policy learning is to derive counterfactual estimators that also has low variance and thus low generalization error.

In this work, inspired by learning bounds for importance sampling problems, we present a new counterfactual learning principle for off-policy learning with bandit feedbacks. Our method regularizes the generalization error by minimizing the distribution divergence between the logging policy and the new policy, and removes the need for iterating through all training samples to compute sample variance regularization in prior work. With neural network policies, our end-to-end training algorithms using variational divergence minimization showed significant improvement over conventional baseline algorithms and is also consistent with our theoretical results.

## 1 Introduction

Off-policy learning refers to evaluating and improving a deterministic policy using historic data collected from a stationary policy, which is important because in real-world scenarios on-policy evaluation is oftentimes expensive and has adverse impacts. For instance, evaluating a new treatment option, a clinical policy, by administering it to patients requires rigorous human clinical trials, in which patients are exposed to risks of serious side effects. As another example, an online advertising A/B testing can incur high cost for advertisers and bring them few gains. Therefore, we need to utilize historic data to perform off-policy evaluation and learning that can enable safe exploration of the hypothesis space of policies before deploying them.

There has been extensive studies on off-policy learning in the context of reinforcement learning and contextual bandits, including various methods such as Q learning (Sutton & Barto (1998)), doubly robust estimator (Dudík et al. (2014)), self-normalized (Swaminathan & Joachims (2015b)), etc. A recently emerging direction of off-policy learning involves the use of logged interaction data with bandit feedback. However, in this setting, we can only observe limited feedback, often in the form of a scalar reward or loss, for every action; a larger amount of information about other possibilities is never revealed, such as what reward we could have obtained had we taken another action, the best action we should have take, and the relationship between the change in policy and the change in reward. For example, after an item is suggested to a user by an online recommendation system, although we can observe the user's subsequent interactions with this particular item, we cannot anticipate the user's reaction to other items that could have been the better options.

Using historic data to perform off-policy learning in bandit feedback case faces a common challenge in counterfactual inference: *How do we handle the distribution mismatch between the logging policy and a new policy and the induced generalization error?* To answer this question, Swaminathan & Joachims (2015a) derived the new counterfactual risk minimization framework, that added the sample variance as a regularization term into conventional empirical risk minimization objective. However, the parametrization of policies in their work as linear stochastic models has limited representation power, and the computation of sample variance regularization requires iterating through

all training samples. Although a first-order approximation technique was proposed in the paper, deriving accurate and efficient end-to-end training algorithms under this framework still remains a challenging task.

Our contribution in this paper is three-fold:

1. By drawing a connection to the generalization error bound of importance sampling (Cortes et al. (2010)), we propose a new learning principle for off-policy learning with bandit feedback. We explicitly regularize the generalization error of the new policy by minimizing the distribution divergence between it and the logging policy. The proposed learning objective automatically trade off between emipircal risk and sample variance.

2. To enable end-to-end training, we propose to parametrize the policy as a neural network, and solves the divergence minimization problem using recent work on variational divergence minimization (Nowozin et al. (2016)) and Gumbel soft-max (Jang et al. (2016)) sampling.

3. Our experiment evaluation on benchmark datasets shows significant improvement in performance over conventional baselines, and case studies also corroborates the soundness of our theoretical proofs.

## 2 BACKGROUND

### 2.1 PROBLEM FRAMEWORK

We first review the framework of off-policy learning with logged bandit feedback introduced in Swaminathan & Joachims (2015a). A policy maps an input $x \in \mathcal{X}$ to a structured (discrete) output $y \in \mathcal{Y}$. For example, the input $x$ can be profiles of users, and we recommend movies of relevance to the users as the output $y$; or in the reinforcement learning setting, the input is the trajectory of the agent, and the output is the action the agent should take in the next time point. We use a family of stochastic policies, where each policy defines a posterior distribution over the output space given the input $x$, parametrized by some $\theta$, i.e., $h_\theta(\mathcal{Y}|x)$. Note that here a distribution which has all its probability density mass on one action corresponds to a deterministic policy. With the distribution $h(\mathcal{Y}|x)$, we take actions by sampling from it, and each action $y$ has a probability of $h(y|x)$ being selected. In the discussion later, we will use $h$ and $h(y|x)$ interchangeably when there will not create any confusion.

In online systems, we observe feedbacks $\delta(x, y; y^*)$ for the action $y$ sampled from $h(\mathcal{Y}|x)$ by comparing it to some underlying 'best' $y^*$ that was not revealed to the system. For example, in recommendation system, we can use a scalar loss function $\delta(x, y; y^*) \to [0, L]$, with smaller values indicating higher satisfaction with recommended items.

The expected risk of a policy $h(\mathcal{Y}|x)$ is defined as

$$R(h) = \mathbb{E}_{x \sim \mathcal{P}(\mathcal{X}), y \sim h(\mathcal{Y}|x)}[\delta(x, y)]$$

, and the goal of off-policy learning is to find a policy with minimum expected risk on test data.

In the off-line logged learning setting, we only have data collected from a logging policy $h_0(\mathcal{Y}|x)$, and we aim to find an improved policy $h(\mathcal{Y}|x)$ that has lower expected risks $R(h) < R(h_0)$. Specifically, the data we will use will be

$$\mathcal{D} = \{x_i, y_i, \delta_i = \delta_i(x_i, y_i), p_i = h_0(y_i|x_i)\}, i = 1, ..., N$$

, where $\delta_i$ and $p_i$ are the observed loss feedback and the logging probability (also called propensity score), and $N$ is the number of training samples.

Two main challenges are associated with this task: 1) If the distribution of a logging policy is skewed towards a specific region of the whole space, and doesn't have support everywhere, feedbacks of certain actions cannot be obtained and improvement for these actions is not possible as a result. 2) since we cannot compute the expectation exactly, we need to resort to empirical estimation using finite samples, which creates generalization error and needs additional regularization.

A vanilla approach to solve the problem is propensity scoring approach using importance sampling (Rosenbaum & Rubin (1983)), by accounting for the distribution mismatch between $h$ and

$h_0$. Specifically, we can rewrite the expected risk w.r.t $h$ as the risk w.r.t $h_0$ using an importance reweighting:

$$
\begin{aligned}
R(h) &= \mathbb{E}_{x \sim \mathbb{P}(\mathcal{X}), y \sim h(y|x)}[\delta(x, y)] \\
&= \mathbb{E}_{x \sim \mathbb{P}(\mathcal{X}), y \sim h_0(y|x)}[\frac{h(y|x)}{h_0(y|x)}\delta(x, y)]
\end{aligned}
\tag{1}
$$

With the collected historic dataset $\mathcal{D}$, we can estimate the empirical risk $\hat{R}_{\mathcal{D}}(h)$, short as $\hat{R}(h)$

$$
\hat{R}(h) = \frac{1}{N}\sum_{i=1}^{N}\frac{h(y_i|x_i)}{h_0(y_i|x_i)}\delta_i(x_i, y_i)
\tag{2}
$$

## 2.2 Counterfactual Risk Minimization

Swaminathan & Joachims (2015a) pointed out several flaws with the vanilla approach, namely, not being invariant to loss scaling, large and potentially unbounded variance. To regularize the variance, the authors proposed a regularization term for sample variance derived from empirical Bernstein bounds.

The modified objective function to minimize is now:

$$
\hat{R}(h) = \frac{1}{N}\sum_{i=1}^{N}u_i + \lambda\sqrt{\frac{Var(\bar{u})}{N}}
\tag{3}
$$

, where $u_i = \frac{h(y_i|x_i)}{h_0(y_i|x_i)}\delta_i$, $\bar{u} = \frac{1}{N}\sum_{i=1}^{N}u_i$ is the average of $\{u_i\}$ obtained from training data, and $Var(\bar{u})$ is the sample variance of $\{u_i\}$.

As the variance term is dependent on the whole dataset, stochastic training is difficult, the authors approximated the regularization term via first-order Taylor expansion and obtained a stochastic optimization algorithm. Despite its simplicity, such first-order approximation neglects the non-linear terms from second-order and above, and introduces approximation errors while trying to reduce the sample variance.

## 3 Variance Regularization Objective

### 3.1 Theoretical Motivation

Instead of estimating variance empirically from the samples, which prohibits direct stochastic training, the fact that we have a parametrized version of the policy $h(\mathcal{Y}|x)$ motivates us to think: can we derive a variance bound directly from the parametrized distribution?

We first note that the empirical risk term $\hat{R}(h)$ is the average loss reweigthed by importance sampling function $\frac{h(y|x)}{h_0(y|x)}$, and a general learning bound exist for importance sampling weights.

Let $z$ be a random variable and the importance sampling weight $w(z) = \frac{p(z)}{p_0(z)}$, where $p$ and $p_0$ are two probability density functions, the following identity holds

**Lemma 1.** *(Cortes et al. (2010)) For a random variable $z$, let $p(z)$ and $p_0(z)$ be two distribuion density function defined for $z$, and $l(z)$ be a loss function of $z$ bounded in $[0, 1]$. Let $w = w(z) = p(z)/p_0(z)$ be the importance sampling weight, the following identity holds:*

$$
\mathbb{E}[w] = 1, \quad \mathbb{E}[w^2] = d_2(p||p_0) = 2^{D_2(p||p_0)}
$$
$$
Var(w) = \mathbb{E}[w^2] - \mathbb{E}[w] = d_2(p||p_0) - 1
$$
$$
\mathbb{E}_{z \sim p_0(z)}[w^2(z)l^2(z)] \leq d_2(p||p_0)
\tag{4}
$$

, *where $D_2$ is the Rényi divergence $D_\alpha$ (Rényi et al. (1961)) with $\alpha = 2$, i.e. squared Chi-2 divergence.*

Based on this lemma, we can derive an upper bound for the second moment of the weighted loss

**Theorem 1.** *Let $X$ be a random variable distributed according to distribution $\mathcal{P}$ with density $p(x)$, $Y$ be a random variable, and $\delta(x, y)$ be a loss function over $(x, y)$ that is bounded in $[0, L]$. For two sampling distributions of y, $h(y|x)$ and $h_0(y|x)$, define their conditional divergence as $d_2(h(y|x)||h_0(y|x); \mathcal{P}(x))$, we have*

$$\mathbb{E}_{x \sim \mathcal{P}(\mathcal{X}), y \sim h_0(y|x)}[w^2(y|x)\delta^2(x, y)] \leq L^2 d_2(h(y|x)||h_0(y|x); \mathcal{P}(x)) \tag{5}$$

The bound is similar to Eq. (4) with the difference that we are now working with a joint distribution over $x, y$. Detailed proofs can be found in Appendix 1.

From the above theorem, we are able to derive a generalization bound between the expected risk $R(h)$ and empirical risk $\hat{R}(h)$ using the distribution divergence function as

**Theorem 2.** *Let $R_h$ be the expected risk of the new policy on loss function $\delta$, and $\hat{R}_h$ be the emprical risk. We additionally assume the divergence is bounded by $M$, i.e., $d_2(h||h_0) \leq d_\infty(h||h_0) = M$*

*Then with probability at least $1 - \eta$,*

$$R(h) \leq \hat{R}(h) + \frac{2LM \log 1/\eta}{3N} + L\sqrt{\frac{2Var(w) \log 1/\eta}{N}}$$

The proof of this theorem is an application of Bernstein inequality and the second moment bound, and detailed proof is in Appendix 7. This result highlights the bias-variance trade-offs as seen in empirical risk minimization (ERM) problems, where $\hat{R}_h$ approximates the emiprcal risk/ bias, and the third term characterize the variance of the solution with distribution divergence (Recall $Var(w) = d_2(h||h_0) - 1$). It thus motivates us that in bandit learning setting, instead of directly optimizing the reweighed loss and suffer huge variance in test setting, we can try to minimize the variance regularized objectives as

$$\min_{h=h(\mathcal{Y}|x)} \hat{R}(h) + \lambda\sqrt{\frac{1}{N}(d_2(h||h_0; \mathcal{P}(x)))) - 1} \tag{6}$$

$\lambda = \sqrt{2L^2 \log 1/\eta}$ is a model hyper-parameter controlling the trade-off between empirical risk and model variance, but we are still faced with the challenge of setting $\lambda$ empirically and the difficuty in optimizing the objective (See Appendix for a comparison). Thus, in light of the recent success of distributionally robust learning, we explore an alternative formulation of the above regularized ERM in the next subsection.

## 3.2 ROBUSTLY REGULARIZED FORMULATION

Instead of solving a 'loss + regularizer' objective function, we here study a closely related constrained optimizationf formulation, whose intuition comes from the method of Langaragian mutliplier for constrained optimization.

The new formulation is:

$$\min_{h} \quad \frac{1}{N}\sum_{i=1}^{m} \frac{h(y_i|x_i)}{p_i}\delta_i$$
$$s.t. \quad d_2(h||h_0; \mathcal{P}(X)) \leq \rho \tag{7}$$

, where $\rho$ is a pre-determined constant as the regularization hyper-parameter.

By applying Theorem , for a policy $h$, we have

$$
\begin{aligned}
R(h) &\leq \hat{R}(h) + \frac{2LM \log 1/\eta}{3N} + L\sqrt{\frac{2Var(w) \log 1/\eta}{N}} \\
&\leq \hat{R}_{d(h||h_0 \leq \rho)}(h) + \frac{2LM \log 1/\eta}{3N} + L\sqrt{\frac{2(\rho - 1) \log 1/\eta}{N}}
\end{aligned}
\tag{8}
$$

This inequality shows that the robust objective $\hat{R}_{d(h||h_0 \leq \rho)}(h)$ is also a good surrogate of the true risk $R(h)$, with their difference bouned by the regularization hyper-parameter $\rho$ and approaches 0 when $N \to \infty$.

At first glance, the new objective function removes the needs to compute the sample variance in existing bounds (3), but when we have a parametrized distribution of $h(y|x)$, and finite samples $\{x_i, y_i\}_{i=1}^N$ from $h_0(y_i|x_i)$, estimating the divergence function is not an easy task. In the next subsection, we will present how recent f-gan networks for variational divergence minimization (Nowozin et al. (2016)) and Gumbel soft-max sampling (Jang et al. (2016)) can help solve the task.

**Discussion: Possibility of Counterfactual Learning:** One interesting aspect of our bounds also stresses the need for the stochasticity of the logging policy (Langford et al. (2008)). For a deterministic logging policy, if the corresponding probability distribution can only have some peaked masses, and zeros elsewhere in its domain, our intution suggests that learning will be difficult, as those regions are never explored. Our theory well reflects this intuition in the calculation of the divergence term, the integral of form $\int_y h^2(y|x)/h_0(y|x)dy$. A deterministic policy has a non-zero measure region of $h_0(\mathcal{Y}|x)$ with probability density of $h_0(y|x) = 0$, while the corresponding $h(y|x)$ can have finite values in the region. The resulting integral results is thus unbounded, and in turn induces an unbounded generalization bound, making counterfactual learning in this case not possible.

## 4 ADVERSARIAL TRAINING ALGORITHM

### 4.1 ADVERSARIAL LEARNING OF THE DIVERGENCE

The derived variance regularized objective (6) requires us to minimize the square root of the conditional divergence, $d_2(h||h_0; \mathcal{P}(X)) = \mathbb{E}_x \int_y \frac{h^2}{h_0} dy$.

For simplicity, we can examine the term inside the expectation operation first. With simple calculation, we have

$$\int_y \frac{h(y|x)^2}{h_0(y|x)} dy = \int_y h_0(y|x)[(\frac{h(y|x)}{h_0(y|x)})^2 - 1 + 1] = D_f(h(\mathcal{Y}|x)||h_0(\mathcal{Y}|x)) + 1$$

, where $f(t) = t^2 - 1$ is a convex function in the domain $\{t : t \geq 0\}$ with $f(1) = 0$. Combining with the expectation operator gives a minimization objective of $D_f(h||h_0; \mathcal{P}(X))$ (+1 omitted as constant).

The above calculation draws connection between our divergence and the f-divergence measure (Nguyen et al. (2010)). Follow the f-GAN for variational divergence minimization method proposed in Nowozin et al. (2016), we can reach a lower bound of the above objective as

$$
\begin{aligned}
D_f(h(\mathcal{Y}|x)||h_0(\mathcal{Y}|x); \mathcal{P}(X)) &= \mathbb{E}_x[\int_y f(\frac{h(y|x)}{h_0(y|x)}) dh_0(y|x)] \\
&= \mathbb{E}_x[\sup_T \{\mathbb{E}_{y \sim h(y|x)}[T(y)] - \mathbb{E}_{y \sim h_0(y|x)}[f^*(T(y))]\}\} \quad (9) \\
&= \sup_T \{\mathbb{E}_x \mathbb{E}_{y \sim h(y|x)}[T(x,y)] - \mathbb{E}_x \mathbb{E}_{y \sim h_0(y|x)}[f^*(T(x,y))]\} \\
&\geq \sup_{T \in \mathcal{T}} \{\mathbb{E}_x \mathbb{E}_{y \sim h(y|x)}[T(x,y)] - \mathbb{E}_x \mathbb{E}_{y \sim h_0(y|x)}[f^*(T(x,y))]\} \\
&= \sup_{T \in \mathcal{T}} \{\mathbb{E}_{x,y \sim h(y|x)} T(x,y) - \mathbb{E}_{x,y \sim h_0(y|x)} f^*(T(x,y))\} \quad (10) \\
&\triangleq F(T, h) \quad (11)
\end{aligned}
$$

For the second equality, as $f$ is a convex function and applying Fenchel convex duality ($f^* = \sup_u \{u'v - f(u)\}$) gives the dual formulation. Because the expectation is taken w.r.t to $x$ while the supreme is taken w.r.t. all functions $T$, we can safely swap the two operators. We note that the bound is tight when $T_0(x) = f'(h/h_0)$, where $f'$ is the first order derivative of $f$ as $f'(t) = 2t$ (Nguyen et al. (2010)).

The third inequality follows because we restrict $T$ to a family of functions instead of all functions. Luckily, the universal approximation theorem of neural networks Hornik et al. (1989) states that neural networks with arbitrary number of hidden units can approximate continous functions on a compact set with any desired precision. Thus, by choosing the family of $T$ to be the family of neural networks, the equality condition of the second equality can be satisfied theoretically.

The final objective (10) is a saddle point of a function $T(x,y) : \mathcal{X} \times \mathcal{Y} \to \mathbb{R}$ that maps input pairs to a scalar value, and the policy we want to learn $h(\mathcal{Y}|x)$ acts as a sampling distribution. Although being a lower bound with achievable equality conditions, theoretically, this saddle point trained with mini-batch estimation is a consistent estimator of the true divergence.

We use $D_f = \sup_T \int T dh dx - \int f^*(T) dh_0 dx$ to denote the true divergence, and $\hat{D}_f \sup_{T \in \mathcal{T}} \int T(x_i, y_i) d\hat{h} dx - \int f^* T(x_j, y_j) d\hat{h}_0 dx$ the empirical estimator we use, where $\hat{h}$ and $\hat{h}_0$ are the emiprical distribution obtained by sampling from the two distribution respectively.

**Propoistion 1.** *$\hat{D}_f$ is a consistent estimator of $D_f$.*

*Proof.* Let's start by decomposing the estimation error. Let $e_0 = (D_f - \sup_{T \in \mathcal{T}} \mathbb{E}[\int T dh - \int f^*(T) dh_0])$, and $e_1 = sup_{T \in \mathcal{T}} |\int T(x,y) d(h - \hat{h}) - \int f^*(T(x,y)) d(h_0 - \hat{h}_0)|$

$$|D_f - \hat{D}_f| \leq e_0 + e_1 \tag{12}$$

, where the first term of error comes from restricting the parametric family of $T$ to a family of neural networks, and the second term of error involves the approximation error of an emipirical mean estimation to the true distribution. By the universal approximation theorem, we have $e_0 = 0$, and that $\exists T \in \mathcal{T}$, such that $T = T_0$.

For the second term $e_1$, we plug in $T_0$ and have it rewritten as

$$sup_{T \in \mathcal{T}} |\int T(x,y) d(h - \hat{h}) - \int f^*(T(x,y)) d(h_0 - \hat{h}_0)| \tag{13}$$

$$\leq |\sup_{T \in \mathcal{T}} [\int (T - T_0)(dh - \hat{dh})] - \int (f^*(T) - f^*(T_0))(dh_0 - \hat{dh}_0)| \tag{14}$$

$$+ |\sup_{T \in \mathcal{T}} T_0(dh - d\hat{h}) - f^*(T_0)(dh_0 - d\hat{h}_0)| \tag{15}$$

For the first term, $|\sup_{T \in \mathcal{T}} T_0(dh - d\hat{h}) - f^*(T_0)(dh_0 - d\hat{h}_0)|$, we can see that this is the diffrence between an empirical distribution and the underlying population distribution. We can verify that the strong law of large numbers (SLLN) applies. By optimality condition, $T_0 = \frac{h(y|x)}{h_0(y|x)}$, where both $h$ and $h_0$ are probability density functions. By the bounded loss assumption, the ratio is integrable. Similarly, $f^*(T_0) = 2T_0 - 1$ is also integrable. Thus, we can apply SLLN and conclude the term $\to 0$. For the second term, we can apply Theroem 5 from [] and also obtain it $\to a.s.0$. $\square$

Again, a generative-adversarial approach (Goodfellow et al. (2014)) can be applied. Toward this end, we represent the $T$ function as a *discriminator* network parametrized as $T_w(x,y)$. We then parametrize the distribution of our policy $h(y|x)$ as another *generator* neural network $h_\theta(y|x)$ mapping $x$ to the probability of sampling $y$. For structured output problems with discrete values of $y$, to allow the gradients of samples obtained from sampling backpropagated to all other parameters, we

use the Gumbel soft-max sampling (Jang et al. (2016)) methods for differential sampling from the distribution $h(y|x)$. We list the complete training procedure Alg. 1 for completeness.

**Data:** $\mathcal{D} = \{x_i, y_i\}_{i=1}^N$ sampled from logging policy $h_0$; a predefined threshold $D_0$; an initial generator distribution $h_{\theta^0}(y|x)$; an initial discriminator function $T_{w^0}(x, y)$
; max iteration $I$ **Result:** An optimized generator $h_{\theta*}(y|x)$ distribution that has minimum divergence to $h_0$
initialization;
**while** $\hat{D}_f(h||h_0; \mathcal{P}(X)) > D_0$ *or iter* $< I$ **do**

    Sample a mini-batch 'real' samples $(x_i, y_i)$ from $\mathcal{D}$ ;

    Sample a mini-batch $x$ from $\mathcal{D}$, and construct 'fake' samples $(x_i, \hat{y}_i)$ by sampling $\hat{y}$ from $h_{\theta^t}(y|x)$ with Gubmel soft-max ;

    Update $w^{t+1} = w^t + \eta_w \partial F(T_w, h_\theta)(10)$ ;

    Update $\theta^{t+1} = \theta^t - \eta_\theta \partial F(T_w, h_\theta)(10)$ ;

**end**

**Algorithm 1:** Variational Minimizing $D_f(h||h_0; \mathcal{P}(X))$

For our purpose of minimizing the variance regularization term, we can similarly derive a training algorithm, as the gradient of $t \to \sqrt{t+1}$ can also be backpropagated.

## 4.2 TRAINING ALGORITHM

With the above two components, we are now ready to present the full treatment of our end-to-end learning for counterfactual risk minimization from logged data. The following algorithm solve the robust regularized formulation and for completeness, training for the original ERM formulation in Sec. 3.1 (referred to co-training version in the later experiment sections) is included in Appendix 7.

**Data:** $\mathcal{D} = \{x_i, y_i, p_i, \delta_i\}_{i=0}^N$ sampled from $h_0$; regularization hyper-parameter $\rho$, and maximum iteration of divergence minimization steps $I$, and max epochs for the whole algorithm $MAX$
**Result:** An optimized generator $h_\theta^*(y|x)$ that is an approximate minimizer of $R(w)$
initialization;
**while** *epoch* $< MAX$ **do**

    `/* Update` $\theta$ `to minimize the reweighted loss                  */`

    Sample a mini-batch of $m$ samples from $\mathcal{D}$ ;

    Estimate the reweighted loss as $\hat{R}^t = \frac{1}{m} \sum_{i=1}^m \frac{h_{\theta^t}(y_i|x_i)}{p_i} \delta_i$ and get the gradient as $g_1 = \partial_\theta R^t$ ;

    Update $\theta^{t+1} = \theta^t - \eta_\theta g_1$ ;

    `/* Update discriminator and generator for divergence`
       `minimization                                              */`

    Call Algorithm 1 to minimize the divergence $D_2(h||h_0; \mathcal{P}(X))$ with threshold $= \rho$, and max iter set to $I$ ;

**end**

**Algorithm 2:** Minimizing Variance Regularized Risk - Separate Training

The algorithm works in two seperate training steps: 1) update the parameters of the policy $h$ to minimize the reweighed loss 2) update the parameters of the policy/ generator and the discriminator to regularize the variance thus to improve the generalization performance of the new policy.

## 5 RELATED WORK

Exploiting historic data is an important problem in multi-armed bandit and its variants such as contextual bandit and has wide applications (Strehl et al. (2010); Shivaswamy & Joachims (2012); Beygelzimer & Langford (2009)). Approaches such as doubly robust estimators (Dudík et al. (2014)) have been proposed, and recent theoretical study explored the finite-time minimax risk lower bound of the problem (Li et al. (2015)), and an adaptive learning algorithm (Wang et al. (2017)) using the theoretical analysis.

Bandits problems can be interpreted as a single-state reinforcement learning (RL) problems, and techniques including doubly robust estimators (Jiang & Li (2015); Thomas & Brunskill (2016); Munos et al. (2016)) have also been extended to RL domains. Conventional techniques such as Q function learning, and temporal difference learning (Sutton & Barto (1998)) are alternatives for off-policy learning in RL by accounting for the Markov property of the decision process. Recent works in deep RL studies have also addressed off-policy updates by methods such as multi-step bootstrapping (Mahmood et al. (2017)), off-policy training of Q functions (Gu et al. (2017)).

Learning from logs traces backs to Horvitz & Thompson (1952) and Rosenbaum & Rubin (1983), where propensity scores are applied to evaluate candidate policies. In statistics, the problem is also described as treatment effect estimation (Imbens (2004)), where the focus is to estimate the effect of an intervention from observational studies that are collected by a different intervention. Bottou et al. (2013) derived unbiased counterfactual estimators to study an example of computational advertising; another set of techniques reduce the bandit learning to a weighted supervised learning problems (Zadrozny et al. (2003)), but is shown to have poor generalization performance (Beygelzimer & Langford (2009)).

Although our variance regularization aims at off-policy learning with bandit feedback, part of the proof comes from the study of generalization bounds in importance sampling problems (Cortes et al. (2010)), where the original purpose was to account for the distribution mismatch between training data and testing distribution, also called covariate shift, in supervised learning. Duchi & Namkoong (2016) also discussed variance regularized empirical risk minimization for supervised learning with a convex objective function, which has connections to distributionally robust optimization problem (Bertsimas et al. (2011)). It will be of further interest to study how our divergence minimization technique can be applied to supervised learning and domain adaptation (Sugiyama et al. (2007); Gretton et al. (2009)) problems as an alternative to address the distribution match issue.

Regularization for our objective function has close connection to the distributionally robust optimization techniques (Bertsimas et al. (2013)), where instead of minizing the emiprical risk to learn a classifier, we minimize the supreme the emipirical risk over an ellipsoid uncertainty set. Wasserstein distance between emipircal distribution and test distribution is one of the most well studied contraint and is proven to achieve robust generalization performance (Esfahani & Kuhn (2015); Gao & Kleywegt (2016)). Distributionally robust optimization is also connected to machine learning, specifically generalization error (Xu et al. (2009); Shafieezadeh-Abadeh et al. (2015)). We refer interested readers to a more comprehensive overview of the subject in Gabrel et al. (2014).

## 6 EXPERIMENTS

### 6.1 EXPERIMENT SETUPS

For empirical evaluation of our proposed algorithms, we follow the conversion from supervised learning to bandit feedback method (Agarwal et al. (2014)). For a given supervised dataset $\mathcal{D}^* = \{(x_i, y_i^*)\}_{i=1}^N$, we first construct a logging policy $h_0(\mathcal{Y}|x)$, and then for each sample $x_i$, we sample a prediction $y_i \sim h_0(y|x_i)$, and collect the feedback as $\delta(y_i^*, y_i)$. For the purpose of benchmarks, we also use the conditional random field (CRF) policy trained on $5\%$ of $\mathcal{D}^*$ as the logging policy $h_0$, and use hamming loss, the number of incorrectly misclassified labels between $y_i$ and $y_i^*$, as the loss function $\delta$ (Swaminathan & Joachims (2015a)). To create bandit feedback datasets $\mathcal{D} = \{x_i, y_i, \delta_i, p_i\}$, each of the samples $x_i$ were passed four times to the logging policy $h_0$ and sampled actions $y_i$ were recorded along with the loss value $\delta_i$ and the propensity score $p_i = h_0(y_i|x_i)$.

In evaluation, we use two type of evaluation metrics for the probabilistic policy $h(\mathcal{Y}|x)$. The first is the expected loss (referred to as 'EXP' later) $R(h) = \frac{1}{N_{test}} \sum_i \mathbb{E}_{y \sim h(y|x_i)} \delta(y_i^*, y)$, a direct measure of the generalization performance of the learned policy. The second is the average hamming loss of maximum a posteriori probability (MAP) prediction $y_{\text{MAP}} = \arg\max h(y|x)$ derived from the learned policy, as MAP is a faster way to generate predictions without the need for sampling in practice. However, since MAP predictions only depend on the regions with highest probability, and doesn't take into account the diverse of predictions, two policies with same MAP performance could have very different generalization performance. Thus, a model with high MAP performance but low EXP performance might be over-fitting, as it may be centering most of its probability masses in the regions where $h_0$ policy obtained good performance.

Table 1: Benchmark Comparison Results.

| Dataset | Scene | | Yeast | | TMC | | LYRL | |
|---|---|---|---|---|---|---|---|---|
| Evaluation Metrics | MAP | EXP | MAP | EXP | MAP | EXP | MAP | EXP |
| Logging Policy $h_0$ (5% data CRF) | 1.069 | 1.887 | 3.255 | 5.485 | 4.995 | 5.053 | 1.047 | 1.949 |
| NN-NoReg | 1.465 | 1.981 | 3.223 | 4.705 | 1.706 | 1.724 | 0.247 | 0.263 |
| NN-Hard | 1.303 | 1.463 | 3.047 | **3.788** | 1.694 | 1.720 | 0.248 | **0.255** |
| NN-Soft | 1.347 | 1.457 | 3.097 | **3.789** | 1.683 | **1.707** | 0.247 | **0.255** |
| IPS | 1.350 | 1.350 | 4.256 | 4.521 | 4.601 | 4.416 | 1.240 | 1.240 |
| POEM | 1.169 | **1.169** | 4.238 | 4.508 | 4.611 | 4.505 | 1.169 | 1.306 |
| IPS(Stochastic) | 1.291 | 1.291 | 4.090 | 4.605 | 2.812 | 2.737 | 1.149 | 1.479 |
| POEM(Stochastic) | 1.322 | 1.323 | 4.140 | 4.570 | 3.601 | 3.561 | 1.237 | 1.237 |
| Supervised Learning (NN) | 0.943 | 2.238 | 3.101 | 4.300 | 1.530 | 3.786 | 0.217 | 0.519 |
| Supervised Learning (CRF) | 1.110 | 1.423 | 2.807 | 4.047 | 1.344 | 1.241 | 0.240 | 0.437 |

## 6.2 BENCHMARK COMPARISON

**Baselines** Vanilla importance sampling algorithms using inverse propensity score (IPS), and the counterfactual risk minimization algorithm from Swaminathan & Joachims (2015a) (POEM) are compared, with both L-BFGS optimization and stochastic optimization solvers. The hyperparameters are selected by performance on validation set and more details of their methods can be found in the original paper (Swaminathan & Joachims (2015a)).

Neural network policies without divergence regularization (short as "NN-NoReg" in later discussions) is also compared as baselines, to verify the effectiveness of variance regularization.

**Dataset** We use four multi-label classification dataset collected in the UCI machine learning repo (Asuncion & Newman (2007)), and perform the supervised to bandit conversion. We report the statistics in Table 2 in the Appendix.

For these datasets, we choose a three-layer feed-forward neural network for our policy distribution, and a two or three layer feed-forward neural network as the discriminator for divergence minimization. Detailed configurations can be found in the Appendix 7.

For benchmark comparison, we use the separate training version 2 as it has faster convergence and better performance (See Sec. 6.5 for an empirical comparison). The networks are trained with Adam (Kingma & Ba (2014)) of learning rate 0.001 and 0.01 respectively for the reweighted loss and the divergence minimization part. We used PyTorch to implement the pipelines and trained networks with Nvidia K80 GPU cards. Codes for reproducing the results as well as preprocessed data can be downloaded with the link [1]

Results by an average of 10 experiment runs are obtained and we report the two evaluation metrics in Table 1. We report the regularized neural network policies with two Gumbel-softmax sampling schemes, soft Gumbel soft-max (NN-Soft), and straight-through Gumbel soft-max (NN-Hard).

As we can see from the result, by introducing a neural network parametrization of the polices, we are able to improve the test performance by a large margin compared to the baseline CRF policies, as the representation power of networks are often reported to be stronger than other models. The introduction of additional variance regularization term (comparing NN-Hard/Soft to NN-NoReg), we can observe an additional improvement in both testing loss and MAP prediction loss. We observe no significant difference between the two Gumbel soft-max sampling schemes.

---

[1] `https://www.dropbox.com/sh/etuc8dnxyope1xh/AAAjFJ06cFyJeYr8YN8z996Ta?dl=0`. We will make them publicly available on GitHub after the anonymous review periods.

## 6.3 EFFECT OF VARIANCE REGULARIZATION

To study the effectiveness of variance regularization quantitatively, we vary the maximum number of iterations ($I$ in Alg. 2) we take in each divergence minimization sub loop. For example, 'NN-Hard-10' indicates that we use ST Gubmel soft-max and set the maximum number of iterations to 10. Here we set the thresholds for divergence slightly larger so maximum iterations are executed so that results are more comparable. We plot the expected loss in test sets against the epochs average over 10 runs with error bars using the dataset yeast.

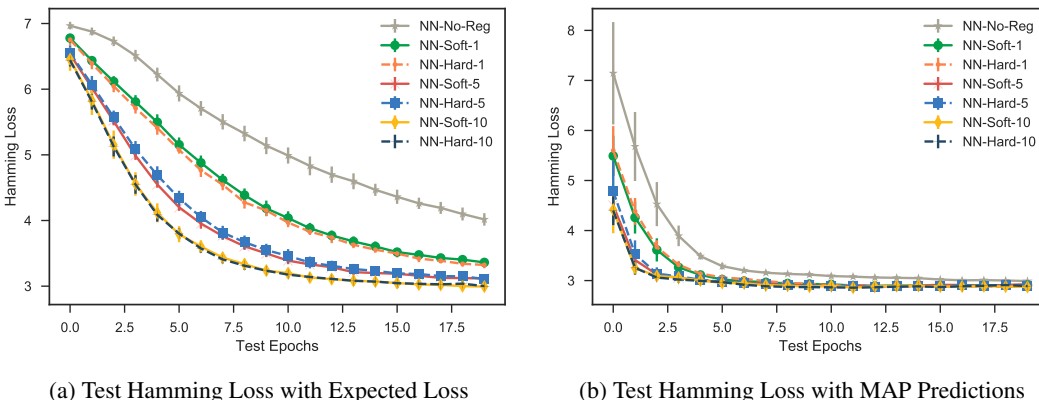

(a) Test Hamming Loss with Expected Loss      (b) Test Hamming Loss with MAP Predictions

Figure 1: Stronger regularization can help obtain faster convergence and better test performance.

As we can see from the figure, models with no regularization (gray lines in the figure) have higher loss, and slower convergence rate. As the number of maximum iterations for divergence minimization increases, the test loss decreased faster and the final test loss is also lower. This behavior suggests that by adding the regularization term, our learned policies are able to generalize better to test sets, and the stronger the regularization we impose by taking more divergence minimization steps, the better the test performance is. The regularization also helps the training algorithm to converge faster, as shown by the trend.

## 6.4 GENERALIZATION PERFORMANCE

Our theoretical bounds implies that the generalization performance of our algorithm improves as the number of training samples increases. We vary the number of passes of training data $x$ was passed to the logging policy to sample an action $y$, and vary it in the range $2^{[1,2,\dots,8]}$ with log scales.

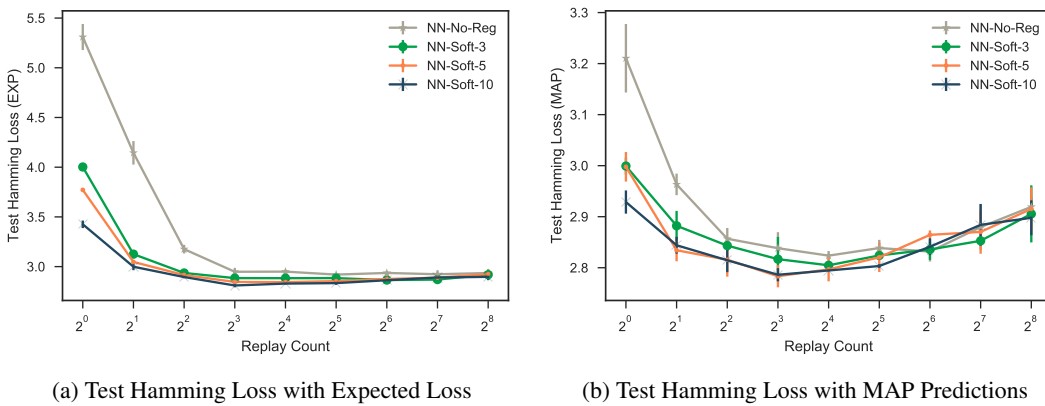

(a) Test Hamming Loss with Expected Loss      (b) Test Hamming Loss with MAP Predictions

Figure 2: Neural network policies both have increasing performance with increasing number of training data, while models with regularization have faster convergence rate and better performance.

When the number of training samples in the bandit dataset increases, both models with and without regularization have an increasing test performance in the expected loss and reaches a relatively stable level in the end. Moreover, regularized policies have a better generalization performance compared to the model without regularization constantly. This matches our theoretical intuitions that explicitly regularizing the variance can help improve the generalization ability, and that stronger regularization induces better generalization performance. But as indicated by the MAP performance, after the replay of training samples are more than $2^4$, MAP prediction performance starts to decrease, which suggests the models may be starting over-fitting already.

## 6.5 TRAINING SCHEMES

In this section, we use some experiments to present the difference in two training schemes: co-training in Alg. 3 and the easier version Alg. 2. For the second algorithm, we also compare the two Gumbel-softmax sampling schemes in addition, denoted as Gumbel-softmax, and Straight-Through (ST) Gumbel-softmax respectively.

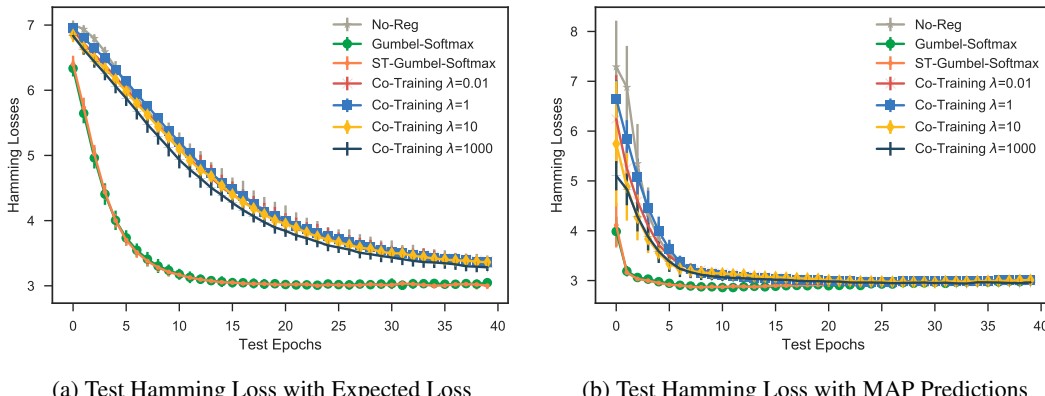

(a) Test Hamming Loss with Expected Loss  (b) Test Hamming Loss with MAP Predictions

Figure 3: Results from different training schemes suggest that separately minimizing reweighted loss and divergence is better compared to training the two losses together.

The figures suggest that blending the weighted loss and distribution divergence performs slightly better than the model without regularization, however, the training is much more difficult compared to the separate training scheme, as it's hard to balance the gradient of the two parts of the objective function. We also observe no significant performance difference between the two sampling schemes of the Gumbel-softmax.

## 6.6 EFFECT OF LOGGING POLICY VS RESULTS

In this section, we discuss how the effect of logging policies, in terms of stochasticity and quality, will affect the learning performance and additional visualizations of other metrics can be found in the Appendix 7.

As discussed before, the ability of our algorithm to learn an improved policy relies on the stochasticity of the logging policy. To test how this stochasticity affects our learning, we modify the parameter of $h_0$ by introducing a temperature multiplier $\alpha$. For CRF logging policies, the prediction is made by normalizing values of $w^T \phi(x, y)$, where $w$ is the model parameter and can be modified by $\alpha$ with $w \to \alpha w$. As $\alpha$ becomes higher, $h_0$ will have a more peaked distribution, and ultimately become a deterministic policy with $\alpha \to \infty$.

We varied $\alpha$ in the range of $2^{[-1,1,\dots,8]}$, and report the average *ratio* of expected test loss to the logging policy loss of our algorithms (Y-axis in Fig 4a, where smaller values indicate a larger improvement). We can see that NN polices are performing better than logging policy when the stochasticity of $h_0$ is sufficient, while after the temperature parameter increases greater than $2^3$, it's much harder and even impossible (ratio ¿ 1) to learn improved NN policies. We also note here that the stochasticity doesn't affect the expected loss values themselves, and the drop in the ratios mainly resulted

from the decreased loss of the logging policy $h_0$. In addition, comparing within NN policies, policies with stronger regularization have slight better performance against models with weaker ones, which in some extent shows the robustness of our learning principle.

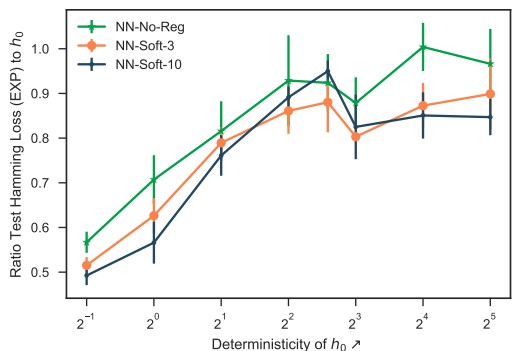
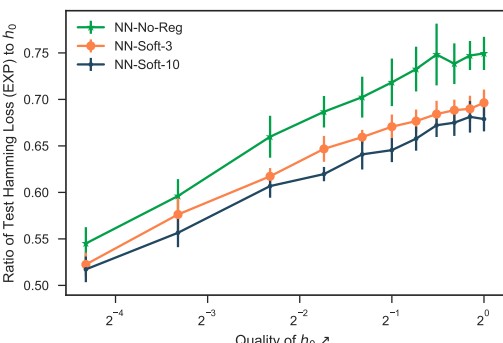

(a) The effect of stochasticity of $h_0$ vs ratio of expected test loss

(b) The effect of quality of $h_0$ vs ratio of expected test loss

Figure 4: a) The decreasing stochasticity of $h_0$ makes it harder to obtain an improved NN policy, and our regularization can help the model be more robust and achieve better generalization performance. b) As $h_0$ improves, the models constantly outperform the baselines, however, the difficulty is increasing with the quality of $h_0$. *Note: more visualizations of other metrics can be found in the appendix 7.*

Finally, we discusses the impact of logging policies to the our learned improved policies. Intuitively, a better policy that has lower hamming loss can produce bandit datasets with more correct predictions, however, it's also possible that the sampling biases introduced by the logging policy is larger, and such that some predictions might not be available for feedbacks. To study the trade-off between better policy accuracy and the sampling biases, we vary the proportion of training data points used to train the logging policy from 0.05 to 1, and compare the performance of our improved policies obtained by in Fig. 4b. We can see that as the logging policy improves gradually, both NN and NN-Reg policies are outperforming the logging policy, indicating that they are able to address the sampling biases. The increasing ratios of test expected loss to $h_0$ performance, as a proxy for relative policy improvement, also matches our intuition that $h_0$ with better quality is harder to beat.

## 7    CONCLUSION

In this paper, we started from an intuition that explicitly regularizing variance can help improve the generalization performance of off-policy learning for logged bandit datasets, and proposed a new training principle inspired by learning bounds for importance sampling problems. The theoretical discussion guided us to a training objective as the combination of importance reweighted loss and a regularization term of distribution divergence measuring the distribution match between the logging policy and the policy we are learning. By applying variational divergence minimization and Gumbel soft-max sampling techniques, we are able to train neural network policies end-to-end to minimize the variance regularized objective. Evaluations on benchmark datasets proved the effectiveness of our learning principle and training algorithm, and further case studies also verified our theoretical discussion.

Limitations of the work mainly lies in the need for the propensity scores (the probability an action is taken by the logging policy), which may not always be available. Learning to estimate propensity scores and plug the estimation into our training framework will increase the applicability of our algorithms. For example, as suggested by Cortes et al. (2010), directly learning importance weights (the ratio between new policy probability to the logging policy probability) has comparable theoretical guarantees, which might be a good extension for the proposed algorithm.

Although the work focuses on off-policy from logged data, the techniques and theorems may be extended to general supervised learning and reinforcement learning. It will be interesting to study how

this training algorithm can work for empirical risk minimization and what generalization bounds it may have as the future direction of research.

## ACKNOWLEDGEMENTS

We thank Dr. Adith Swaminathan for publishing their codes and answering the authors' questions. We'd also like to thank XXX, YYY, ZZZ for their precious comments and TTT for their funding support.

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

## A. PROOFS

PROOF FOR THEOREM 1

*Proof.* Let $z = (x, y)$, $p(z) = \mathcal{P}(x)h(y|x)$, $p_0(z) = \mathcal{P}(x)h_0(y|x)$, and $l(z) = \delta(x, y) \in [0, L]$. We apply Lemma 1 to $z$, importance sampling weight function $w(z) = p(z)/p_0(z) = h(y|x)/h_0(y|x)$, and loss $l(z)/L$, we have

$$
\begin{aligned}
\mathbb{E}_{x \sim \mathcal{P}(\mathcal{X}), y \sim h_0(y|x)}[w^2(y|x)\delta^2(x,y)/L^2] &= \mathbb{E}_{z \sim p_0(z)}[w^2(z)l^2(z)/L^2] \\
&\leq d_2(p(z)||p_0(z)) \\
&= \int_z \frac{p(z)}{p_0(z)} p(z) dz \\
&= \int_{x,y} \frac{h(y|x)}{h_0(y|x)} h(y|x)\mathcal{P}(x) dx dy \\
&= \int_x d_2(h(y|x)||h_0(y|x))\mathcal{P}(x) dx \\
&= d_2(h(y|x)||h_0(y|x); \mathcal{P}(x))
\end{aligned}
$$

Thus, we have

$$
\mathbb{E}_{z \sim p_0(z)}[w^2(z)l^2(z)] \leq L^2 d_2(p||p_0)
$$

$\square$

PROOF FOR THEOREM 3.2

*Proof.* For a single hypothesis denoted as $\delta$ with values $\delta_i = \delta(z_i)$, let $Z = w_h(z)l(z) - R(h)$, then $|Z| \leq LM$. By Lemma 1, the variance can be bounded using Reni divergence as

$$
Var(Z) = \mathbb{E}_{z \sim p_0(z)}[w^2(z)l^2(z)] - R_h^2 \leq L^2 d_2(p||p_0) = R_h^2
$$

Applying Bernstein's concentration bounds we have

$$
Pr(R(h_0) - \hat{R}(h) > \epsilon) \leq \exp(\frac{-m\epsilon^2/2}{\sigma^2(Z) + \epsilon LM/3})
$$

Let $\eta = \exp(\frac{-m\epsilon^2/2}{\sigma^2(Z) + \epsilon LM/3})$, we can obtain that with probability at least $1 - \eta$, the following bounds for importance sampling of bandit learning holds

$$
\begin{aligned}
R_{h_0} &\leq \hat{R}_h + \frac{LM \log 1/\eta}{3m} + \sqrt{\frac{L^2 M^2 \log^2 1/\eta}{9m^2} + \frac{2L^2 Var(Z) \log 1/\eta}{m}} \\
&\leq \hat{R}_h + \frac{2LM \log 1/\eta}{3m} + \sqrt{\frac{2L^2 Var(Z) \log 1/\eta}{m}}
\end{aligned}
\tag{16}
$$

, where the second inequality comes from the fact that $\sqrt{a+b} \leq \sqrt{a} + \sqrt{b}$. $\square$

TRAINING ALGORITHM FOR ORIGINAL ERM FORMULATION

**Data:** $\mathcal{D} = \{x_i, y_i\}_{i=1}^{N}$ sampled from logging policy $h_0$; regularization hyper-parameter $\lambda$
**Result:** An optimized generator $h_\theta^*(y|x)$ that is an approximate minimizer of $R(w)$
initialization;
**while** *Not Converged* **do**
> /* Update discriminator                                                 */
> Sample a mini-batch of 'fake' samples $(x_i, \hat{y}_i)$ with $x_i$ from $\mathcal{D}$ and $\hat{y}_i \sim h_{\theta^t}(y|x_i)$;
> Sample a mini-batch of 'real' samples $(x_i, y_i)$ from $\mathcal{D}$ ;
> Update $w^{t+1} = w^t + \eta_w \partial F(T_w, h_\theta)(10)$ ;
> /* Update generator                                                     */
> Sample a mini-batch of $m$ samples from $\mathcal{D}$ ;
> Estimate the reweighted loss as $\hat{R}^t = \frac{1}{m} \sum\limits_{i=1}^{m} \frac{h_{\theta^t}(y_i|x_i)}{p_i} \delta_i$ and get the gradient as $g_1 = \partial_\theta R^t$ ;
> Sample a mini-batch of $m_1$ 'fake' samples ;
> Estimate the generator gradient as $g_2 = F(T_w, h_\theta)(10)$ ;
> Update $\theta^{t+1} = \theta^t - \eta_\theta(g_1 + \lambda g_2)$ ;

**end**

**Algorithm 3:** Minimizing Variance Regularized Risk - Co-Training Version

## A. STATISTICS OF BENCHMARK DATASETS

We report the statistics of the datasets as in the following table. For the latter two datasets TMC,

Table 2: Dataset Statistics

| Name | # Features | # Labels | # Train | # Test |
|------|-----------|----------|---------|--------|
| Yeast | 103 | 14 | 1500 | 917 |
| Scene | 294 | 6 | 1211 | 1196 |
| TMC | 30438 | 22 | 21519 | 7077 |
| LYRL | 47236 | 4 | 23149 | 781265 |

LYRL, as they have sparse features with high dimension of features, we first reduced their feature dimensions to 1000 via truncated singular value decomposition (latent semantic analysis).

## B. NETWORK STRUCTURES

For policy networks, we use the following simple network structure

```
Linear -> BatchNorm -> ReLU -> Linear -> BatchNorm
-> ReLU -> Linear -> Sigmoid
```

, mainly because of features used in the datasets don't invole imaging or original text features. For the discriminator, a three-layer design is

```
Linear -> BatchNorm -> ReLU -> Linear -> BatchNorm -> ReLU -> Linear
```

## C. SUPPLEMENT FIGURES TO SECTION 6.6

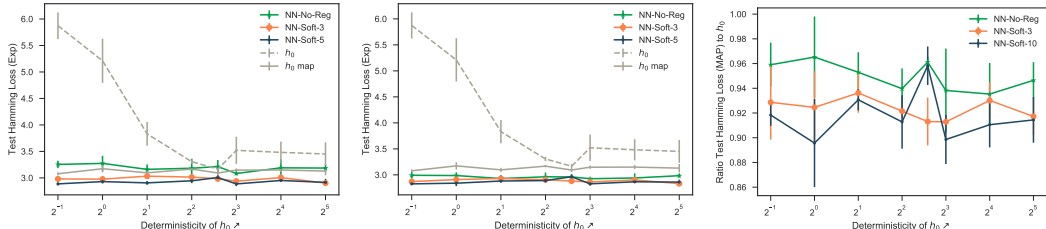

(a) The effect of stochasticity of $h_0$ (b) The effect of stochasticity of $h_0$ (c) The effect of stochasticity of $h_0$
vs expected test loss     vs test loss with MAP predictions     vs ratio of test loss with MAP

Figure 5: As the logging policy becomes more deterministic, NN policies are still able to find improvement over $h_0$ in a) expected loss and b) loss with MAP predictions. c) We cannot observe a clear trend in terms of the performance of MAP predictions. We hypothesize it results from that $h_0$ policy already has good MAP prediction performance by centering some of the masses. While NN policies can easily pick up the patterns, it will be difficult to beat the baselines. We believe this phenomenon worth further investigation.

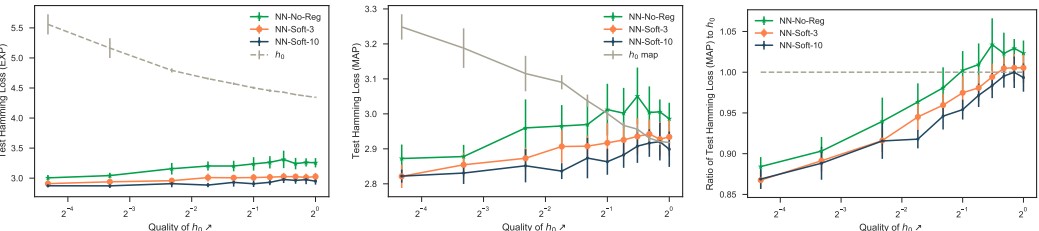

(a) The quality of $h_0$ vs ratio of ex-(b) The quality of $h_0$ vs ratio of ex-(c) The quality of $h_0$ vs ratio of ex-
pected test loss     pected test loss     pected test loss with MAP

Figure 6: a) As the quality of the logging policy increases, NN policies are still able to find improvement over $h_0$ in expected loss. and b) c) For MAP predictions, however, it will be really difficult for NN policies to beat if the logging policy was already exposed to full training data and trained in a supervised fashion.

