# OpenReview forum: "Variance Regularized Counterfactual Risk Minimization via Variational Divergence Minimization"
_ICLR.cc/2018/Conference — Reject_

### Official Review · AnonReviewer3 · 2017-11-26
**Variance Regularized Counterfactual Risk Minimization via Variational Divergence Minimization**

**Rating:** 4
**Confidence:** 4

**Review:**

In this paper the authors studied the problem of off-policy learning, in the bandit setting when a batch log of data generated by the baseline policy is given. Here they first summarize the surrogate objective functions derived by existing approaches such as importance sampling and variance regularization (Swaminathan et. al). Then they extend the results in Theorem 2 of the paper by Cortes et. al (which also uses the empirical Bernstein inequality by Maurer and Pontil), and derive a new surrogate objective function that involves the chi-square divergence. Furthermore, the authors also show that the lower bound of this objective function can be iteratively approximated by variational f-GAN techniques, which could potentially be more numerically stable and empirically has lower variance.

In general, I think the problem studied in this paper is very interesting, and the topic of counterfactual learning, especially policy optimization with the use of offline and off-policy log data, is important. However, I think the theoretical contribution in this paper on off-policy learning is quite incremental. Also the parts that involve f-GAN is still questionable to me.

Detailed comments:
In these variance regularization formulations (for example the one proposed in this paper, or the one derived in Swaminathan's paper), \lambda can be seen as a regularization parameter that trades-off bias and variance of the off-policy value estimator R(h) (for example the RHS of equation 6). To exactly calculate \lambda either requires the size of the policy class (when the policy class is finite), or the complexity constants (which exists in C_1 and C_2 in equation 7, but it is not clearly defined in this paper). Then the main question is on how to choose \lambda such that the surrogate objective function is reasonable. For example in the safety setting (off-policy policy learning with baseline performance guarantees, for example see the problem setting in the paper by P. Thomas 2015: High Confidence off-policy improvement), one always needs the upper-bound in 6) to hold. This makes the choice of \lambda crucial and challenging. Unfortunately I don't see much discussions in this paper about choosing \lambda, even in the context of bias-variance trade-offs. This makes me uncomfortable in believing that the results in experiments hold for other (reasonable) choices of \lambda.

The contribution of this paper is of two-fold: 1) the authors extend the results from Cortes's paper to derive a new surrogate objective function, and 2) they show how this objective can be approximated by f-GAN techniques. The first contribution is rather incremental as it's just a direct application of Theorem 2 in Cortes's paper. Regarding the second contribution, I am a bit concerned about the derivations of Equation 9, especially the first inequality and the second equality. I see that the first inequality is potentially an application of the conjugate function inequality, but more details are needed (f^* is not even defined). For the second equality, it's unclear to me how one can swap the sup and the E_x operators. More explanations are definitely needed to show their mathematical correctness, especially when this part is a main contribution.  Even if the derivations are right, the f-GAN surrogate objective is a lower bound of the surrogate objective function, while the surrogate function is an upper bound of the true objective function (which is inaccessible). How does one guarantees that the f-GAN surrogate objective is a reasonable one?

Numerical comparisons between the proposed approach, and the approach from Swaminathan's paper are required to demonstrate the superiority of the proposed approach. Are there comparisons in performance between the approach from the original chi-square surrogate function and the one from the f-GAN objective (in order to showcase the need of using f-GAN) as well?

Minor comments:
In experimental section, method POEM is not defined.
The paper is in an okay status. But there are several minor typos, for example \hat{R}_{(} in page 3, and several typos in Algorithm 1 and Algorithm 2.

In general, I think this paper is studying an interesting topic, but the aforementioned issues make me feel that the paper's current status is still unsuitable for publication.

---

> ### Author Response · Authors · 2018-01-05
> **Author response to "Variance Regularized Counterfactual Risk Minimization via Variational Divergence Minimization"**
>
> Dear reviewer,
>
> Thanks a lot for your valuable comments and we have revised our manuscript and hope it can address some of your concerns.
>
> Below are our point-by-point correspondence.
>
> - (choosing \lambda & bias-variance trade-offs) We agree with you this is indeed a very difficult question, as with other regularized ERM techniques.
>
> As our Theorem 2 suggests, \lambda is \sqrt{L^2 log(1/\eta) / N}, where L is the bound of the loss, \eta is the probability of the bound to hold, and N is the number of samples, this sheds some light to how the \lambda controls the bias-variance trade-off, where true risk < ERM + \sqrt{(divergence-1)/N} + O(1).
>
> The bound here is mainly controlled by the divergence between the new policy h and the empirical log policy h0, which implicitly includes the sample variance.
>
> In ERM, we have E_h[loss(classifier, label)], where we have a distribution h, and classifier, both coming from two families.
> Here the loss is already fixed and provided by dataset, and the concentration bound exists because of the variance of importance sampling bounds of h/h0 (Theorem 2 and its proof), and doesn't concern the hypothesis space from losses/ classifiers.
>
> In practice however, we believe the best approach is still start with the intuition from the theorem and play with cross validation.
>
> - (lower bound vs upper bound)  We apologize for the confusion here.
>
> the lower bound exists because we are restricting the family of discriminators from all functions to all neural networks. This leads to the lower bound, however, since neural networks are essentially universal function approximators [1], the equality condition can be satisfied here in theory.
>
> For the first and second inequality, the first one is an application of Fenchel duality and we also found the equality condition can be satisfied here. The swapping of sup T and E_x works because T is actually only a function of y so can be left out for integration over x.
>
> We have updated our proofs and discussion and hope we can make things clearer.
>
> - (numerical comparisons) The POEM algorithm is the method from the
> Adith et al paper, and we believe the results demonstrated the superior performance of our approach.
>
> The chi-square surrogate function (NN-CoTraining: min loss + \lambda divergence ) comparison with the f-gan objective (NN-SeparateTraining, min loss s.t. divergence \leq \rho) can be found in Sec. 6.5.
>
> For separate training, chi-square and f-gan has no difference because it only has difference in the RHS constraint by a constant 1. For Co-training, we have found the two approaches don't work well, while separate training performs much better than co-training
>
> Thanks a lot for your insightful comments and please let us know if you need further clarifications from us!
>
> Best,
> Author
>
> [1] Hornik, Kurt, Maxwell Stinchcombe, and Halbert White. "Multilayer feedforward networks are universal approximators." Neural networks 2.5 (1989): 359-366.

---

### Official Review · AnonReviewer1 · 2017-11-26
**Interesting combination of off-policy learning from bandit feedback and f-GANs, with some weaknesses in theory and experiment**

**Rating:** 5
**Confidence:** 5

**Review:**

The paper proposes an interesting alternative to recent approaches to learning from logged bandit feedback, and validates their contribution in a reasonable experimental comparison. The clarity of writing can be improved (several typos in the manuscript, notation used before defining, missing words, poorly formatted citations, etc.).
Implementing the approach using recent f-GANs is an interesting contribution and may spur follow-up work. There are several lingering concerns about the approach (detailed below) that detract from the quality of their contributions.

[Major] In Lemma 1, L(z) is used before defining it. Crucially, additional assumptions on L(z) are necessary (e.g. |L(z)| <= 1 for all z. If not, a trivial counter-example is: set L(z) >> 1 for all z and Lemma 1 is violated). It is unclear how crucially this additional assumption is required in practice (their expts with Hamming losses clearly do not satisfy such an assumption).

[Minor] Typo: Section 3.2, first equation; the integral equals D_f(...) + 1 (not -1).

[Crucial!] Eqn10: Expected some justification on why it is fruitful to *lower-bound* the divergence term, which contributes to an *upper-bound* on the true risk.

[Crucial!] Algorithm1: How is the condition of the while loop checked in a tractable manner?

[Minor] Typos: Initilization -> Initialization, Varitional -> Variational

[Major] Expected an additional "baseline" in the expts -- Supervised but with the neural net policy architecture (NN approaches outperforming Supervised on LYRL dataset was baffling before realizing that Supervised is implemented using a linear CRF).

[Major] Is there any guidance for picking the new regularization hyper-parameters (or at least, a sensible range for them)?

[Minor] The derived bounds depend on M, an a priori upper bound on the Renyi divergence between the logging policy and any new policy. It's unclear that such a bound  can be tractably guessed (in contrast, prior work uses an upper bound on the importance weight -- which is simply 1/(Min action selection prob. by logging policy) ).

---

> ### Author Response · Authors · 2018-01-05
> **Author response to "Interesting combination of off-policy learning from bandit feedback and f-GANs, with some weaknesses in theory and experiment"**
>
> Dear reviewer,
>
> Thanks a lot for the inspiring comments and below are our point-by-point correspondence and hope the revision can address these concerns and make the paper more solid.
>
> - (Citations formatting) We have fixed the missing parenthesis for end-of-sentence citations. We apologize for the inconvenience because mistaking the hyper-ref box for actual parenthesis.
>
> - ( Z and loss L(z)) You are absolutely right the loss needs to be bounded in [0,1] for the theorem from Cortes et al. to be valid. In this work, because we didn't perform loss scaling to [0,1] but instead assume the loss [0,L]. We revised our theorems to reflect such bounded loss condition.
>
> As for experiments, since the loss is only scaling factor, the minimization problem and its solution is essentially the same ( \sum loss vs. \sum loss/L).
>
> - (typos for +1 instead of -1) fixed
>
> - (why lower bound works) We apologize for the confusion here.
>
> the lower bound exists because we are restricting the family of discriminators from all functions to all neural networks. This leads to the lower bound, however, since neural networks are essentially universal function approximators [1], the equality condition can be satisfied here in theory.
>
> For the first and second inequality, the first one is an application of Fenchel duality and we also found the equality condition can be satisfied here. The swapping of sup T and E_x works because T is actually only a function of y so can be left out for integration over x.
>
> We have updated our proofs and discussion and hope we can make things clearer.
>
> - (loop condition check) We use the estimator of divergence function obtained from empirical distributions as an approximation and check the value against the threshold.
>
> Furthermore, we have also established through proof that, the estimator using empirical distribution is a consistent estimator of the true divergence. (Sec. 4.1 proposition 1)
>
> - (typos) fixed
>
> - (baselines) why logging policy is CRF instead of NN? As our latter experiment shows, as the logging policy gets better, it will be more difficult to improve upon the logging policy, esp. IPS POEM which uses linear CRF as policy. so we only used linear CRF as the logging policy.
>
> (EDIT 1:50PM, 1/5: We just realized we have misunderstood your intent was not to suggest using NN as logging policy but show the performance of NN trained with supervised methods. We are really sorry but it's approaching the rebuttal deadline, we hope to provide such statistics asap)
>
> (EDIT : We just found out it is still possible to upload a modified manuscript, so we went ahead to upload an updated table. We used the exact architecture as of the NN policy with bandit training. The NN learned with supervised training has similar performance compared to CRF, and we think it might result from the over-fitting, as indicated by the high EXP loss)
>
> - (picking regularization hyper-parameter) This is indeed a very difficult question, as with other regularized ERM techniques.
>
> As our Theorem 2 suggests, \lambda is \sqrt{L^2 log(1/\eta) / N}, where L is the bound of the loss, \eta is the probability of the bound to hold, and N is the number of samples, this sheds some light to how the \lambda controls the bias-variance trade-off, where true risk < ERM + \sqrt{(divergence-1)/N} + O(1).
>
> In practice however, we believe the best approach is still start with the intuition from the theorem and play with cross validation.
>
> - ( " it's unclear that such a bound can be tractably guessed ") We have provided a full proof of the bound, which is essentially an application of Bernstein inequality and Theorem 1 of our paper. But please let us know if you feel anything is unclear.
>
> Again, we thanks for all the great comments you made to help us improve the paper and we really appreciate it.
>
> Best,
> Authors
>
> [1] Hornik, Kurt, Maxwell Stinchcombe, and Halbert White. "Multilayer feedforward networks are universal approximators." Neural networks 2.5 (1989): 359-366.

---

### Official Review · AnonReviewer2 · 2017-11-27
**Well written paper, good contribution by leveraging several diverse work but average/limited applicability**

**Rating:** 7
**Confidence:** 3

**Review:**

This paper studies off-policy learning in the bandit setting. It develops a new learning objective where the empirical risk is regularized by the squared Chi-2 divergence between the new and old policy. This objective is motivated by a bound on the empirical risk, where this divergence appears. The authors propose to solve this objective by using generative adversarial networks for variational divergence minimization (f-GAN). The algorithm is then evaluated on settings derived from supervised learning tasks and compared to other algorithms.

I find the paper well written and clear. I like that the proposed method is both supported by theory and empirical results.

Minor point: I do not really agree with the discussion on the impact of the stochasticity of the logging policy in section 5.6. Based on Figure 5 a and b, it seems that the learned policy is performing equally well no matter how stochastic the logging policy is. So I find it a bit misleading to suggest that the learned policy are not being improved when the logging policy is more deterministic. Rather, the gap reduces between the two policies because the logging policy gets better. In order to better showcase this mechanism, perhaps you could try using a logging policy that does not favor the best action.

quality and clarity:
++ code made available
+ well written and clear
- The proof of theorem 2 is not in the paper nor appendix (the authors say it is similar to another work).


originality
+ good extension of the work by Swaminathan & Joachims (2015a): derivation of an alternative objective and use of a deep networks
. This paper leverages a set of diverse results

significance
- The proposed method can only be applied if propensity scores were recorded when the data was generated.
- no test on a real setting
++ The proposed method is supported both by theoretical insights and empirical experiments.
+ empirical improvement with respect to previous methods


details/typos:

3.1, p3: R^(h) has an indexed parenthesis
5.2; and we more details
5.3: so that results more comparable

---

> ### Author Response · Authors · 2018-01-05
> **Author response to "Well written paper, good contribution by leveraging several diverse work but average/limited applicability"**
>
> Dear reviewer,
>
> Thanks a lot for the insightful comments! Below are our point-by-point correspondence to your reviews and hope they can address some of your concerns and clarify things a bit.
>
> - (Regarding the stochasticity vs performance). Thanks for the suggestion for using a different type of policy and we experimented with policies with random exploration, i.e., \hat p = (1-\epsilon) p + \epsilon, where p is the NN policy output, and similar trends hold.
>
> As you may have noticed in the figure, the y values, ratio = test performance / logging policy performance gets greater than 1 after the stochasticity is considerably small. This suggests that the learned policy performs worse than logging policy on average, and demonstrates our main point in this experiment: as long as there are enough stochasticity in the policy, it is possible to learn an improved policy, while it is really hard to learn from a very deterministic policy.
>
> We have revised the paragraph and hopefully it won't create further confusion.
>
> - (The proof of theorem 2) We've added the proof in the appendix section.
>
> - (Need for propensity scores) Yes, the availability of propensity scores is crucial to the algorithm. Cortes et al. provided a way to learn propensity scores and showed similar generalization bound, and we think this will be a very interesting future work. But the scope of this paper limits our exploration in this direction.
>
> - (No test on a real setting) We agree this is one of the limitation of our algorithm, but deploying such algorithm on-line such as large-scale ads placement system, is something we don't have access right now, so we have to resort to simulation studies.
>
> - (typos) Fixed them and thanks for pointing out!
>
> We really appreciate your positive feedbacks on our paper overall and please let us know if our response needs further explanation.
>
> Thanks again for your effort in reviewing our paper!
>
> Best,
> Authors

---

> > ### Comment · AnonReviewer2 · 2018-01-12
> > **Final thoughts**
> >
> > Thank you for your reply and for updating the paper.
> >
> > Regarding the stochasticity, I might misunderstand Figure 5 a, so let me clarify my comment. Figure 5a seems to contain the values used to compute Figure 4a. If this is correct, then the decrease in improvement shown in Figure 4a is not due to a worse learned policy (due to stochasticity) but to the improvement of h0.
> >
> > In other words, the learned policies in Figure 5a have approximately constant performance no matter the stochasticity of h0. To me, this suggests the stochasticity of h0 does not impact the performance of the learned policy.

---

> > > ### Author Response · Authors · 2018-01-16
> > > **Absolute vs Relative Performance**
> > >
> > > Dear Reviewer,
> > >
> > > Thanks a lot for the follow-up and clarification. You're correct about Fig. 4a) and 5a) and that the absolute loss doesn't change too much w.r.t to the stochasticity .
> > >
> > > In the paper, when we were trying to discuss how logging policy affected the final performance, we used the relative performance (in our mind), performance(h)/performance(h0), to indicate how easy it is to improve upon h0.
> > >
> > > In the stochastic experiment, based on the relative performance, we concluded it was harder and harder to improve upon h0, as h0 became more deterministic. However, the ability to improve upon h0 doesn't necessarily mean model performance, and our writings confused with two concepts. We've updated our writings to make a clear contrast.
> > >
> > > Another point we were trying to make in the experiment is that NN policies without regularization performed slightly worse than the ones with regularization, which we think reflects the benefit of the proposed regularization. The trend is more observable in Fig. 4a) because of the scale of the Y-axis, so that we kept the figures the way they were.
> > >
> > > Thanks again for pointing this out and the help in improving our manuscript!
> > >
> > > Best,
> > > Authors

---

### Author Response · Authors · 2018-01-05
**Revision Update**

As pointed out by reviewers, major concerns they had is the theoretical soundness of the paper, so we updated our derivation and proofs, and below is the summary of the major revision:

1) by leveraging Bernstein inequality and our lemma of bounding the second moment of importance sampling weights, we reached a regularized erm formulation as
          True Risk < Empirical Risk + \lambda \sqrt{ divergence(new policy| logging policy)} /N + C/N
, which highlights the bias-variance trade-offs.

\lambda is \sqrt{L^2 log(1/\eta) / N}, where L is the bound of the loss, \eta is the probability of the bound to hold, and N is the number of samples.

2) with intuition from constrained optimization, we proposed a modified regularization as:
         min Empirical Risk s.t. divergence(new policy| logging policy) \leq \rho
and this is shown to be a good surrogate of the true loss

3) for computing the divergence(new policy| logging policy), the major concern is that we reached a lower bound of the divergence with adversarial training, while we are minimizing an upper bound.

we apologize for the confusion here, the lower bound comes from restricting the family of discriminators from all functions to neural networks. Theoretically, because of the universal approximation ability of neural networks, the equality condition can be satisfied.

we also added a proof showing that the empirical estimator we are minimizing is a consistent estimator of the true divergence

4) other minor re-organizations and small fixes such as typos

---

### Decision · Program_Chairs · 2018-01-29
**ICLR 2018 Conference Acceptance Decision**

**Decision:**

Reject

**Comment:**

The reviewers agree that the paper studies and interesting problem with an interesting approach. The reviewers raised some concerns regarding the theoretical and empirical results. The authors have made changes to the paper, but given the theoretical nature of the paper and the extent of changes, another review is needed before publication.